# Musashi-1 Is a Novel Immunohistochemical Marker of Neuroendocrine Carcinoma of the Lung

**DOI:** 10.3390/cancers15235631

**Published:** 2023-11-29

**Authors:** Yu Izaki, Vishwa Jeet Amatya, Takahiro Kambara, Kei Kushitani, Yoshihiro Miyata, Morihito Okada, Yukio Takeshima

**Affiliations:** 1Department of Surgical Oncology, Research Institute for Radiation Biology and Medicine, Hiroshima University, Hiroshima 734-8551, Japan; blaugrana.jp@gmail.com (Y.I.); ymiyata@hiroshima-u.ac.jp (Y.M.); 2Department of Pathology, Graduate School of Biomedical and Health Sciences, Hiroshima University, Hiroshima 734-8551, Japan; amatya@hiroshima-u.ac.jp (V.J.A.); kambara0213@gmail.com (T.K.); kkushi@hiroshima-u.ac.jp (K.K.); ykotake@hiroshima-u.ac.jp (Y.T.)

**Keywords:** pathological diagnosis, Musashi-1, small cell carcinoma, large cell neuroendocrine carcinoma, lung cancer, surgical specimen, novel marker, immunohistochemistry

## Abstract

**Simple Summary:**

Small cell lung cancer (SCLC) and large cell neuroendocrine carcinoma (LCNEC) of the lung are now classified as lung neuroendocrine carcinomas (NECs) and have different treatment strategies compared with non-SCLC. Although these NECs are less frequently resected by surgery, the diagnosis of LCNEC is particularly difficult in some cases because staining for neuroendocrine markers is necessary in addition to morphological features. In our retrospective study of pathology specimens from 42 surgically resected SCLCs and 44 LCNECs, we demonstrated that immunohistochemical staining with Musashi-1, a marker of neural stem cells, more diffusely and intensely stained SCLC and LCNEC tissues. The findings of this study may improve and simplify the diagnosis of NECs in surgical and biopsy specimens and increase the opportunity to provide more appropriate treatment for patients.

**Abstract:**

Small cell lung cancer (SCLC) and large cell neuroendocrine carcinoma (LCNEC) have recently been grouped as lung neuroendocrine carcinomas (NECs). Because these lung NECs are clinically malignant and their treatment strategies differ from those of non-SCLC, the quality of diagnosis has a significant prognostic impact. The diagnosis of LCNEC requires positive immunohistochemical staining with chromogranin A, synaptophysin, and CD56, along with a morphological diagnosis, and insulinoma-associated protein 1 (INSM1) has been proposed as an additional marker but is still not an ideal or better marker. We investigated Musashi-1 as a novel immunohistochemical marker in 42 patients with SCLCs and 44 with LCNECs who underwent lung resection between 1998 and 2020 at our institution. We found Musashi-1 expression in 98% (41/42) SCLC and in 90% (40/44) LCNEC. These findings were similar to CD56 expression and superior to synaptophysin, chromogranin A, and INSM1. Musashi-1 also tended to show more diffuse and intense staining, especially in LCNEC, with more cases staining > 10% than any other existing markers (Musashi-1, 77%; INSM1, 45%; chromogranin A, 34%; synaptophysin, 41%; and CD56, 66%). In conclusion, we identified Musashi-1 as a novel immunohistochemical staining marker to aid in the diagnosis of lung NEC.

## 1. Introduction

Lung cancer is the leading cause of cancer deaths in Japan and worldwide, first among men and second among women [1]. The prognosis and treatment guidelines for patients with lung cancer differ significantly between neuroendocrine carcinoma (NEC), primarily small cell lung cancer (SCLC), and non-small cell lung cancer (NSCLC) [2]. SCLCs account for approximately 15% of all newly diagnosed lung cancers and have a poor prognosis due to its extremely high proliferative activity and early metastasis to lymph nodes and systemic sites [3]. According to the 2015 World Health Organization (WHO) Classification of Lung Tumors, large cell neuroendocrine carcinoma (LCNEC) was removed from the category of large cell carcinoma and grouped with SCLC and carcinoid tumors as the neuroendocrine tumors of the lung. LCNEC and SCLC were classified as high-grade neuroendocrine tumors of the lung for their poorly differentiated features [4].

SCLCs show distinct morphological, ultrastructural, and immunohistochemical characteristics. The characteristics of SCLC include densely packed small tumor cells with scant cytoplasm and a poorly defined cell border forming sheet-like diffuse growth patterns, however, neuroendocrine features, such as nesting, trabecular, peripheral palisading, and rosette formations, may be present. SCLC may present as pure small cell carcinoma or combined with other carcinoma components, such as adenocarcinoma, squamous cell carcinoma, and LCNEC [4]. Although immunohistochemistry is not always necessary for its diagnosis, the identification of neuroendocrine differentiation is helpful for the identification of SCLC or component of SCLC in combined SCLC. Immunohistochemical demonstration of neuroendocrine differentiation using chromogranin A, synaptophysin, and CD56, in conjunction with morphological findings, significantly improves the reproducibility of the SCLC diagnosis [5]. However, 10% of SCLCs are negative for all three commonly used neuroendocrine markers. LCNEC has also been diagnosed as an NEC using similar immunostaining. Furthermore, positive findings in relation to either immunohistochemical stain are required for LCNEC in the current pathological diagnosis [4]. Because these NECs are clinically more malignant and progress more rapidly than NSCLCs and the treatment strategies used for these cancers differ from those for NSCLCs, a rapid and correct diagnosis in biopsy or surgically removed lung cancer tissue has a significant impact on prognosis [6].

Recently, insulinoma-associated protein 1 (INSM1) has been proposed as an additional and better marker of NEC, including SCLC [7,8,9]. However, further studies from multiple institutions about its application have shown the pitfall of its utility despite its high sensitivity and specificity for NEC [10].

Therefore, we focused on Musashi-1 as a new immunohistochemical marker to further aid and simplify diagnosis. Musashi-1 is a 39-kDa ribonucleic acid binding protein. It was originally reported to be associated with maintenance and asymmetric cell division of neural and epithelial progenitor cells. It is selectively expressed in neural progenitor cells, including neural stem cells [11,12], and is also identified as a stem cell marker of glioma [13], colorectal cancer [14], and gastric cancer [15]. Because NEC of the lungs is also poorly differentiated and has neuroendocrine characteristics, it is likely that Musashi-1 is expressed as in the other cancers already reported earlier. Although there have been reports of Musashi-1 expression in all histologic types of lung cancer [16], it remains to be revealed whether Musashi-1 is useful in NEC.

In this study, we found that Musashi-1 is strongly and diffusely expressed in LCNEC and SCLC. We also investigated its usefulness as a novel immunohistochemical marker for NEC in clinical pathology by comparing it with commonly used existing neuroendocrine markers.

## 2. Materials and Methods

### 2.1. Gene Expression Data Analysis from the Cancer Dependency Map

A cancer dependency map (https://depmap.org/portal/ (accessed on 14 September 2023)) was used to analyze gene expressions in SCLC and NSCLC cell lines. The gene expression levels (Expression Public 23Q2) were compared between SCLC and NSCLC cell lines using the “two-class comparison” analysis function. Volcano plots were generated for genes with effect sizes of less than −2 or greater than 2. Scatter plots were generated for the Musashi-1 expression levels, which were the largest effect size (Figure 1).

### 2.2. Patients and Histological Samples

The pathology specimens of patients with SCLC and LCNEC who underwent lung resection at Hiroshima University Hospital from 1998 to December 2020 were retrieved from the archives of the Department of Surgical Oncology, Hiroshima University, and 42 SCLC and 44 LCNEC cases were listed. Three pathologists (V.J.A., K.K., and Y.T.) have carefully examined all hematoxylin and eosin-stained histological sections, and one of the best representative sections was further analyzed for immunohistochemical staining with chromogranin A, synaptophysin, CD56, and INSM1. Immunohistochemistry with TTF-1, Napsin-A, CEA, Claudin 4, P40, and CK5/6 was performed to reclassify according to the WHO histological classification. Forty-two SCLCs and forty-four LCNECs were examined for the expression of Musashi-1 and other neuroendocrine markers (chromogranin A, synaptophysin, CD56, and INSM1). The same immunohistochemical expression was also examined in 80 NSCLC cases, and the false-positive rate was determined in each marker. This study was conducted according to the Ethics Guidelines for Human Genome/Gene Research enacted by the Japanese Government for the collection of tissue specimens and approved by the Institutional Ethics Review Committee (Hiroshima University E-2078). 

### 2.3. Immunohistochemical Procedures and Evaluation of Musashi-1 Expression

Immunohistochemistry was performed using 3 µm tissue sections prepared from the best representative formalin-fixed paraffin-embedded tissue blocks. All immunohistochemical staining was performed using the Ventana BenchMark GX automated immunohistochemical station (Roche Diagnostics, Tokyo, Japan). The details of the antibody and conditions of the immunohistochemical procedures are summarized in Table 1. For immunohistochemistry for Musashi-1, antigen retrieval was carried with Ventana CC1 buffer (Ventana, Roche Diagnostics, Tokyo, Japan) for 30 min followed by incubation with the anti-Musashi-1 antibody (Rabbit monoclonal antibody EP1302, at 1:250 dilution; Abcam, Inc., Cambridge, UK). Incubation with the secondary antibody and detection were performed with the Ventana ultraView Universal DAB Detection Kit (Roche Diagnostics), and Mayer’s hematoxylin solution was used for nuclear staining. Immunoreactivity was interpreted as either negative (no immunostaining) or positive by assessing the expression in the cytoplasm of the tumor cells. Furthermore, positive immunoreactivity was scored as 0 for none, 1+ for up to 10%, 2+ for >10% to 50%, and 3+ for >50% of tumor cells showing immunohistochemical expression. Only the NEC component was evaluated in combined SCLC or combined LCNEC of the lung.

Using the Human Protein Atlas database (https://www.proteinatlas.org/ (accessed on 14 September 2023)), we searched for organs that do not stain with Musashi-1 and confirmed that normal lung tissue does not stain. The absence of staining of normal lung tissue in pathology specimens served as a negative control for the antibody.

## 3. Results

### 3.1. Immunohistochemical Results of NEC

The percentages of positive cases and their immunohistochemical scores of Musashi-1 and other neuroendocrine markers—chromogranin A, synaptophysin, CD56, and INSM1—in NEC are summarized in Table 2. 

Musashi-1 expression was primarily observed in the cytoplasm of NEC. Musashi-1 expression was observed in 41 SCLCs (98%) and 40 LCNECs (91%). In SCLC, thirty-four patients had an immunohistochemical score of 3+, six had a score of 2+, and one had a score of 1+. In LCNEC, twenty-nine patients had an immunohistochemical score of 3+, six had a score of 2+, and six had a score of 1+. The tissue specimens of representative cases stained with Musashi-1 are shown in Figure 2.

Forty SCLCs (95%) and thirty-one LCNECs (68%) expressed INSM1. The immunohistochemical scores were 3+ in thirty-four SCLCs and fourteen LCNECs, 2+ in two SCLCs and six LCNECs, and 1+ in four SCLCs and ten LCNECs, showing a trend toward lower expression, especially in LCNEC, than Musashi-1. Chromogranin A, synaptophysin, and CD56 were expressed in 32, 37, and 39 SCLC cases and 23, 29, and 40 LCNEC cases, respectively. 

### 3.2. Immunohistochemical Results of NSCLC

Musashi-1 expression was observed in 14 NSCLCs (18%). In the majority of cases, expression was found only in a small region of the tumor cells; however, one patient had an immunohistochemical score of 3+, and three had a score of 2+. Chromogranin A, synaptophysin, CD56, and INSM1 were expressed in three (4%), fifteen (19%), fifteen (19%), and ten (13%) NSCLC cases, respectively. The detailed immunohistochemical scores are shown in Table 2.

### 3.3. Results with a Cutoff of >10% Staining for Each Marker

When the cutoff line was set to a staining rate of at least 10%, Musashi-1 was found to be expressed in forty SCLCs (95%) and thiry-four LCNECs (77%), with four cases of staining in NSCLC with a specificity of 95%. For existing markers, chromogranin A stained 67% of SCLCs and 34% of LCNECs; synaptophysin, 73% and 41%; CD56, 95% and 66%; and INSM1, 86% and 45%. All of these stains were less sensitive than Musashi-1; however, specificity was similar in all cases (Table 3). 

### 3.4. Venn Diagram Showing the Number of Duplicate Staining Cases

The Venn diagram shows the overlap of the stained cases of Musashi-1 and the three neuroendocrine markers currently used in the routine practice of pulmonary NEC at our institution for a total of 86 NEC cases, including 42 SCLC and 44 LCNEC cases (Figure 3). Cases with >1% staining are shown in Figure 3A, and those with >11% staining are shown in Figure 3B.

## 4. Discussion

In general, pulmonary NEC is more aggressive and has a poorer prognosis than NSCLC, and treatment strategies, such as surgical indications and techniques and chemotherapy drugs, are different. LCNEC, previously distinguished from SCLC, has recently been reported as a cancer with a poorer prognosis than other NSCLC [17,18]. Because the time spent on chemotherapy that is not sensitive to the cancer being treated and the misjudgment of whether surgery is indicated can clearly worsen a patient’s prognosis, it is extremely important to improve diagnostic quality before the start of cancer treatment in daily clinical practice [6]. Although SCLC is less advanced compared with NSCLC in terms of recent development of targeted therapies, the diagnosis of NEC is expected to become more important in the future, as new targeted therapies may emerge based on the reports of genetic analysis and classification [19,20]. In addition, it is desirable to start cancer treatment at an earlier point in time, and simplifying diagnosis is an equally important issue in today’s clinical practice, where a large number of cancer cases of various types are being diagnosed and treated. 

The diagnosis of SCLC can be reliably made based on routine histological and cytological preparations; however, immunohistochemistry may be required for the confirmation of the neuroendocrine and epithelial nature of the tumor cells [5]. In addition to morphological diagnosis, staining for immunohistological markers is essential for the diagnosis of LCNEC. A panel of neuroendocrine markers, including CD56, chromogranin A, and synaptophysin, will improve the diagnostic accuracy of these NECs [4,21]. Moreover, the identification of a new neuroendocrine marker, INSM1, has been reported to be a better marker than these prior-established neuroendocrine markers [7,8,9]. The sensitivity rates of INSM1 and CD56 are better than those of chromogranin A and synaptophysin. The present study also found that the sensitivity rates of INSM1 and CD56 are higher than those of synaptophysin and chromogranin A. Recently, the sensitivity rates of chromogranin A and synaptophysin have also significantly improved, and this may be considered because of the automation of the immunohistochemical technique and use of improved antibody clones. In this study, the sensitivity of INSM1 (95%) was similar to those of synaptophysin (90%) and CD56 (95%) in SCLC but was considerably higher than that of chromogranin (83%). In LCNEC, the sensitivity of CD56 (89%) was higher than those of INSM1 (68%) and synaptophysin (66%) but lower than that of chromogranin A (52%). These results are generally similar to those of recent studies, which have reported that INSM1 is useful in NECs of the lung, whereas others have reported low sensitivity [10]. In addition, the fifth edition of the WHO Classification of Neuroendocrine Cancer states that further study is needed [4]. The sensitivity of INSM1 in LCNEC in the present study was lower than those in reports recommending immunohistochemical staining with this neuroendocrine marker [7,8,9]. However, these reports on INSM1 were based on fewer cases of LCNEC than in the present study, and the sensitivity ranged from 75% to 100%; therefore, the sensitivity of this marker in LCNEC may not be high enough to be a diagnostic marker for LCNEC [7,8,9,22].

On the other hand, the sensitivity of immunohistochemical staining with Musashi-1 was better than those with other neuroendocrine markers for SCLC (98%) and LCNEC (91%). To the best of our knowledge, this is the first study to analyze Musashi-1 expression in SCLC and LCNEC. Musashi-1 was originally reported to be associated with maintenance and asymmetric cell division of neural and epithelial progenitor cells [11,12,23]. The expression of Musashi-1 in normal glial cells of the cerebrum and cerebellum, retinal cells, and glioma [13,24] led us to hypothesize the possibility of Musashi-1 expression in NEC of the lung. Although there have been previous reports that Musashi-1 is associated with lung cancer [16], the results of Musasi-1 localization in the CNS and expression analysis in the cancer dependency map as described above support our contention that Musasi-1 has a stronger correlation with lung NEC.

We found Musashi-1 expression in all 42 SCLC cases and 40 out of 44 LCNEC cases. In addition, Musashi-1 expression in SCLC was strong and diffuse, with only one case showing an immunohistochemical score of 1+. In LCNEC, a higher percentage showed 2+ or 3+ staining than any other traditional markers. On the other hand, the frequency of Musashi-1 expression in NSCLC was equal to or greater than that of conventional neuroendocrine markers; however, in most cases, the staining was focal and weak.

Especially for LCNEC, for which proof of staining of immunohistochemical markers is essential for diagnosis, diffuse and strong staining may simplify the diagnosis in routine practice and prevent oversights. Because this was a retrospective study, the fact that LCNEC was diagnosable in surgical specimens and the fact that any of the immunohistochemical markers were stained were synonymous, and only one SCLC case was found to be negative for any of the markers. Although it would be ideal to demonstrate the usefulness of Musashi-1 in cases where LCNEC is strongly suspected morphologically but conventional immunohistochemical markers are negative, this has not been possible in this study. Instead, we demonstrated that Musashi-1 stained more cases extensively than existing markers. To demonstrate its usefulness in real-world clinical practice, the number of cases overlapping in Musashi-1 with the three immunohistochemical markers used in the actual diagnosis of the cases in this study is presented visually. As shown in the Venn diagram in Figure 3, 13 of the NEC specimens that stained no more than 10% with all existing markers used at the time of clinical diagnosis showed >10% staining with Musashi-1. This suggests that if Musashi-1 had been used at the time of clinical diagnosis, diagnostic work could have been performed more smoothly. In addition, Musashi-1 staining was obtained in five of the NEC cases with no staining at all for CD56, which has excellent sensitivity among existing immunohistochemical markers, and Musashi-1 staining was obtained in 17 cases when limited to cases with >10% staining for CD56. In other words, the stained cases did not overlap with existing markers, suggesting the possibility of expanding the diagnostic range when used in combination.

The wide range of staining with Musashi-1 also has the potential to be effective even when tumor cells in the specimen are small. In addition to surgical biopsy, transbronchial lung biopsy and computed-tomography-guided lung biopsy are commonly used in clinical practice to diagnose lung cancer. These nonsurgical lung biopsies are often difficult to diagnose because of inadequate tissue volume, and it is a common experience in clinical lung resections that the diagnoses of preoperative and postoperative histopathological types differ [25]. Especially in LCNEC, preoperative diagnosis has been reported to be extremely difficult, and the wide range of immunohistochemical staining may improve diagnostic quality in such situations where the amount of tissue is limited and may be of more benefit in combined SCLC and combined LCNEC [26,27,28].

The specificity of Musashi-1 for lung NEC tested with adenocarcinoma and squamous cell carcinoma (81%) was comparable to those of synaptophysin (82%) and CD56 (81%) and lower than those of chromogranin (96%) and INSM1 (87%). Musashi-1 tended to stain more diffusely than existing markers but with lower specificity. Therefore, when the cutoff line was set to >10% staining (the percentage of staining described as necessary for diagnosis with a single marker in the fourth edition of the WHO classification), the sensitivity was not reduced compared with those of the existing markers, and the specificity of the marker was higher than those of the existing markers (Table 3). Musashi-1 is important to diagnose in combination with other markers because of the need for caution when the staining area is slight; however, when the staining is more diffuse, Musashi-1 alone may be useful for pathological diagnosis.

As mentioned earlier, Musashi-1 was also stained in NSCLC in several cases, and its percentage was comparable to or higher than those of existing neuroendocrine markers. On the other hand, it has been reported that neuroendocrine differentiation is observed in 10–20% of NSCLCs [4,29]. The clinical significance of this finding has not been proven, and it is not officially recognized as a tumor classification in the current WHO classification. Musashi-1 staining in this study was found in 18% of NSCLCs, which is comparable to the frequency of neuroendocrine differentiation found in NSCLCs in the aforementioned report. The Musashi-1-positive cases in this study are summarized in Table 4. No significantly higher recurrence rates or other findings were observed. To further investigate these associations, future studies should use more NSCLC cases, especially those with follow-up of postoperative recurrence and post-recurrence histology to examine the association between Musashi-1 expression and neuroendocrine differentiation in NSCLC.

On the other hand, some existing neuroendocrine markers, such as INSM1 and Nestin, have been reported to be associated with the prognosis of NEC [30,31]. Although in this study no staining was performed on the specimens, POU2F3 has also recently been implicated as a subtype of NEC as well as a possible prognostic factor and useful as an immunohistochemical marker in pathology [32,33]. As mentioned earlier, the development of novel therapies such as driver gene-targeted therapy is significantly less developed in NEC compared to NSCLC, and reports of novel markers have the potential to contribute to improved prognosis not only as prognostic predictors but also as therapeutic targets. If Musashi-1 is proven to be ubiquitously expressed in NEC in the future, it may be necessary to consider its potential as a prognostic predictor and therapeutic target.

Despite the encouraging results, our findings do not provide compelling evidence to justify complete replacement of traditional neuroendocrine markers with Musashi-1. First, a limitation of this study was the small number of cases because it was based on lung surgery specimens with sufficient tissue volume. In recent clinical practice, with the start of treatment with immune checkpoint inhibitors and other new drugs and the advent of gene panel testing, it has been necessary to store sufficient amounts of cancer tissue specimens, especially from surviving patients. There were some cases in which it was difficult to obtain tissue for research purposes. The numbers of SCLC and LCNEC cases were small to begin with, and the number of cases for which surgery was indicated was also limited, making it difficult to use a large number of cases at a single institution. To improve the accuracy of the study, analysis in collaboration with other institutions is being considered.

In SCLC, traditional markers are easy to interpret in most cases. Musashi-1 would be of great value if it was positive in SCLCs that were negative for all three traditional neuroendocrine markers. However, we did not encounter a single instance of this situation in the present study. Furthermore, because only cases that met the traditional diagnostic criteria for LCNEC (extensive staining for a single marker or staining for two or more markers) could be picked up from the database, it was not possible to determine whether Musashi-1 would be useful in diagnosing LCNECs that were negative for any of the traditional markers. However, Musashi-1 stained more extensively for LCNEC than either of the traditional markers in this study, with a greater frequency of 2+ or higher, suggesting that it is extremely useful in cases where staining with other markers is limited and the diagnosis is difficult to make. Figure 2 shows two cases of extensive staining with Musashi-1 among cases that were difficult to diagnose because of the extremely limited staining with existing markers.

In conclusion, Musashi-1 expression is a novel marker for NEC of the lung, which may be combined with traditional markers to further simplify the diagnosis.

## 5. Conclusions

This study suggests that immunohistochemical staining with Musashi-1 enhances the diagnostic sensitivity of NECs. In addition, Musashi-1 tended to stain a wider range of NEC tissues than the existing markers. Although more in-depth studies are needed in the future, it may serve as a more convenient diagnostic aid for SCLC and LCNEC as novel markers of lung NEC.

## Figures and Tables

**Figure 1 cancers-15-05631-f001:**
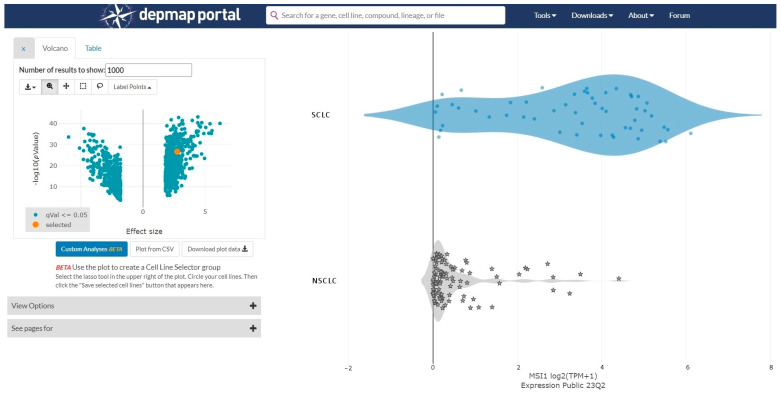
A cancer dependency map (https://depmap.org/portal/ (accessed on 14 September 2023)) is used to analyze gene expressions in small cell lung cancer (SCLC) and non-small cell lung cancer (NSCLC) cell lines. The gene expression levels are compared between SCLC and NSCLC cell lines. The effect size is 2.76 and the *p*-value is 2.30 × 10^27^.

**Figure 2 cancers-15-05631-f002:**
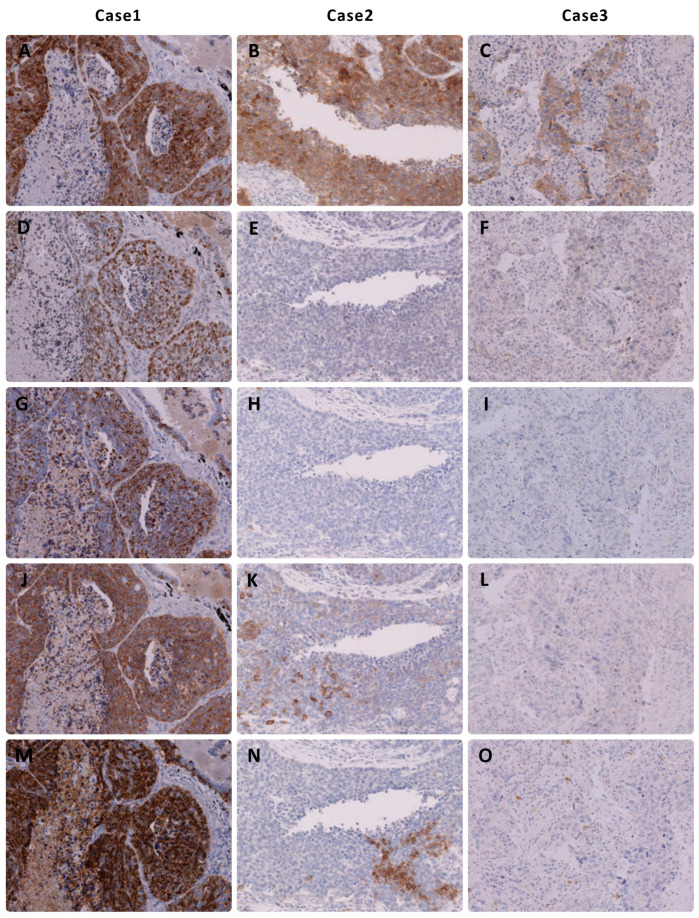
Immunohistochemical findings of representative cases of large cell neuroendocrine carcinoma of the lung. Note that (**A**–**C**) Musashi-1 expression shows a high immunohistochemical score in all three cases. (**D**–**F**) Insulinoma-associated protein 1, (**G**–**I**) chromogranin A, and (**J**–**L**) synaptophysin show no or low expression in two cases. (**M**–**O**) CD56 is not expressed in Figure (**O**).

**Figure 3 cancers-15-05631-f003:**
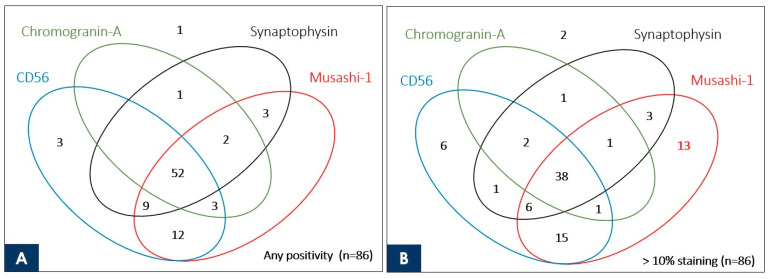
Venn diagram showing the correlation of commonly used neuroendocrine markers and Musashi-1 with the number of stained cases for 86 neuroendocrine carcinomas (44 large cell neuroendocrine carcinomas and 42 small cell lung cancers). The number indicated inside the oval is the case stained with each marker. Although there are no cases in which Musashi-1 is stained alone (**A**), Musashi-1 stains alone > 10% in 13 neuroendocrine carcinoma cases, which is better than the existing marker (**B**).

**Table 1 cancers-15-05631-t001:** Antibody and source of neuroendocrine markers of small cell carcinoma of the lung.

Antibody to	Clone	Source	Dilution	Antigen Retrieval
Musashi-1	EP-1302	Abcam	1:250	CC1 30 min
INSM1	C-8	Santa Cruz	1:100	CC1 60 min
Chromogranin-A	LK2H10	Ventana-Roche	prediluted	CC1 60 min
Synaptophysin	MRQ-40	Ventana-Roche	prediluted	CC1 60 min
CD56	MRQ-42	Ventana-Roche	prediluted	CC1 30 min

INSM1 = insulinoma-associated protein 1.

**Table 2 cancers-15-05631-t002:** Immunohistochemical expression of Musashi-1 and other neuroendocrine markers in lung cancer.

**SCLC**	**Total Cases**	**Number of Positive Cases**	**Immunohistochemical Score**			
			**0**	**1+**	**2+**	**3+**
Musashi-1	42	41 (98%)	1	1	6	34
INSM1	42	40 (95%)	2	4	2	34
Chromogranin-A	42	35 (83%)	7	7	6	22
Synaptophysin	42	38 (90%)	4	7	0	31
CD56	42	40 (95%)	2	0	9	31
**LCNEC**	**Total Cases**	**Number of Positive Cases**	**Immunohistochemical Score**			
			**0**	**1+**	**2+**	**3+**
Musashi-1	44	40 (91%)	4	6	6	28
INSM1	44	30 (68%)	14	10	6	14
Chromogranin-A	44	23 (52%)	21	8	1	14
Synaptophysin	44	29 (66%)	15	11	1	17
CD56	44	39 (89%)	5	10	6	23
**NSCLC**	**Total Cases**	**Number of Positive Cases**	**Immunohistochemical Score**			
			**0**	**1+**	**2+**	**3+**
Musashi-1	80	14 (18%)	66	10	3	1
INSM1	80	10 (13%)	70	10	0	0
Chromogranin-A	80	3 (4%)	77	3	0	0
Synaptophysin	80	15 (19%)	66	10	4	1
CD56	80	15 (19%)	65	11	4	0

SCLC = small cell lung cancer, LCNEC = large cell neuroendocrine carcinoma, NSCLC = non-small cell lung cancer, INSM1 = insulinoma-associated protein-1.

**Table 3 cancers-15-05631-t003:** Sensitivity and specificity in any positive case and >10% of stained cases.

	Any Positivity			>10% Positive Cells		
	Sensitivity		Specificity	Sensitivity		Specificity
	SCLC *n* = 42	LCNEC *n* = 44	NSCLC *n* = 80	SCLC *n* = 42	LCNEC *n* = 44	NSCLC *n* = 80
Musashi-1	41 (98%)	40 (91%)	66 (83%)	40 (95%)	34 (77%)	76 (95%)
INSM1	40 (95%)	30 (68%)	70 (87%)	36 (86%)	20 (45%)	80 (100%)
Chromogranin-A	35 (83%)	23 (52%)	77 (96%)	28 (67%)	15 (34%)	80 (100%)
Synaptophysin	38 (90%)	29 (66%)	65 (81%)	31 (73%)	18 (41%)	75 (94%)
CD56	40 (95%)	39 (89%)	65 (81%)	40 (95%)	29 (66%)	76 (95%)

SCLC = small cell lung cancer, LCNEC = large cell neuroendocrine carcinoma, INSM1 = insulinoma-associated protein-1.

**Table 4 cancers-15-05631-t004:** Clinical background of cases of non-small cell lung cancer stained with MSI-1.

Case	MSI-1	Syn	CD56	ChrA	ISNM1	Type	pStage	Smoke (BI)	Rec
1	3+	0	2	0	1	Ad (pap)	IA2	+(1600)	-
2	2+	0	0	0	0	Sq	IA2	+(1000)	-
3	2+	0	0	0	1	Sq	IB	+(860)	-
4	2+	1	0	0	0	Sq	IB	+(600)	-
5	1+	0	0	0	1	Sq	IIB	+(3000)	-
6	1+	0	2	0	0	Sq	IIIB	+(2700)	+
7	1+	0	1	0	0	Sq	IB	+(1920)	-
8	1+	0	1	0	1	Sq	IB	+(1800)	-
9	1+	1	0	0	0	Sq	IIIA	+(1380)	+
10	1+	0	0	0	1	Ad (aci)	IB	+(1320)	-
11	1+	0	1	0	0	Sq	IB	+(1200)	+
12	1+	0	0	1	1	Ad (pap)	IB	+(1075)	-
13	1+	0	0	0	0	Sq	IB	+(660)	-
14	1+	0	0	0	0	Sq	IA2	+(600)	-

MSI-1 = Musashi-1, Syn = synaptophysin, ChrA = chromogranin-A, INSM1 = insulinoma-associated protein-1, pStage = pathological stage, BI = Brinkman index, rec = recurrence, Ad = adenocarcinoma, Sq = squamous cell carcinoma, pap = papillary predominant, aci = acinar predominant.

## Data Availability

The data that support the findings of this study are available from the corresponding author, [Takeshima Y], upon reasonable request.

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
