# Peer review of "Musashi-1 Is a Novel Immunohistochemical Marker of Neuroendocrine Carcinoma of the Lung"

_cancers, 2023, doi:10.3390/cancers15235631_

Round 1

Reviewer 1 Report

Comments and Suggestions for Authors

The manuscript entitled " Musashi-1 Is a Novel Immunohistochemical Marker of Neuroendocrine Carcinoma of the Lung" highlights the use of Musashi-1 as a novel diagnostic marker for neuroendocrine carcinomas. The authors do a fantastic job in explaining the rationale behind their research work and show good evidence supporting their claims. However, few revisions may be need before accepting the paper for publication - 

1. It would be greatly benefit the authors to give stronger justifications for the selecting Musashi-1 from their gene expression analysis. It might be prudent for mentioning fold change expression of musashi-1 along with p-value.

2. It would be prudent to cite this article: https://www.ncbi.nlm.nih.gov/pmc/articles/PMC3742834/ because the idea of Musashi as a potential therapeutic / diagnostic marker for lung cancer has been suggested. 

Author Response

Comments 1: It would be greatly benefit the authors to give stronger justifications for the selecting Musashi-1 from their gene expression analysis. It might be prudent for mentioning fold change expression of musashi-1 along with p-value.

[response]

Thank you for giving us the opportunity to make our presentation even better. We have added the effect size and P-value you mentioned to the description of Figure1. (The effect size is 2.76 and the P-value is 2.30e-27)

Comment 2: It would be prudent to cite this article: https://www.ncbi.nlm.nih.gov/pmc/articles/PMC3742834/ because the idea of Musashi as a potential therapeutic / diagnostic marker for lung cancer has been suggested.

[response]

Thank you for the very useful references. We have cited the relevant paper as reference 16 and added it to the background in the introduction (lines 83-85) and in the discussion (lines 244-247).

Reviewer 2 Report

Comments and Suggestions for Authors

Dear Editor, 

The manuscipt "Musashi-1 Is a Novel Immunohistochemical Marker of Neuroendocrine Carcinoma of the Lung"by Izaki Y et al is very interesting paper about expression of Musashi-1, a novel IHC marker for neuroendocrine carcinoma of the lung.

The study is well done, and the paper well written, including limitations and possible further studies.

I have just a few suggestions/questions which if possible to address, could improve the manuscript.

1. Main objection is that the authors have not mentioned another recently described and important marker- POU2F3. it would be great to have the results of staining also for this one.

Minor issues:

1. In Simple summary- I would rephrase/change words in the following sentence......may improve the diagnostic quality and simplicity of NECs.....here simplicity might be not quite understandable...maybe to rephrase as ...may improve and simplify diagnosis of NECs

2. In Abstract, ...please rephrase..(INSM1) has been proposed as an additional marker BUT SOMETIMES EXPERIENCE DIFFICULTY in making diagnosis...NOw it looks like marker itself has some problems...maybe just to change it with...but still not ideal or still not much better marker...

3. Also in Abstract...please change Musashi-1 expression in 41 (98%) of 42 SCLC to in 98% (41/42) SCLC....and the same for LCNEC

4. In Introduction, please add also that lung cancer is the leading cause of death in JApan and Worldwide!

5. In introduction, line 69-70...a rapid and highly diagnostic quality...should be rephrased...maybe change to ...a rapid and correct diagnosis in biopsy or surgically removed....

6. In the introduction, line 84, please change to...In this study, we found that Musashi-1 is strongly and diffusely expressed in LCNEC and SCLC.

7. Last words in the description of Figure 2...is it correct Figure 1???

8. In Figure 3, I would suggest the full name and not MSI-1, or adding a legend?

Kind regards

Author Response

  1. Main objection is that the authors have not mentioned another recently described and important marker- POU2F3. it would be great to have the results of staining also for this one.

[response]

We appreciate your comments and the opportunity to clarify this important point.

The report of POU2F3 in lung cancer is mentioned in lines 307-314 of the main text and cites references 32 and 33.

Several of the lung cancer specimens used in this study are difficult to obtain at this time because they are still alive and will be needed for future treatment or because the amount of tissue is small. It is difficult to prepare and stain new specimens.

Minor issues:

  1. In Simple summary- I would rephrase/change words in the following sentence......may improve the diagnostic quality and simplicity of NECs.....here simplicity might be not quite understandable...maybe to rephrase as ...may improve and simplify diagnosis of NECs

[response]

Thank you for pointing this out. We have changed the relevant part to the appropriate text.

  1. In Abstract, ...please rephrase..(INSM1) has been proposed as an additional marker BUT SOMETIMES EXPERIENCE DIFFICULTY in making diagnosis...NOw it looks like marker itself has some problems...maybe just to change it with...but still not ideal or still not much better marker...

[response]

Thank you for pointing this out. We have changed the relevant part to the appropriate text.

  1. Also in Abstract...please change Musashi-1 expression in 41 (98%) of 42 SCLC to in 98% (41/42) SCLC....and the same for LCNEC

[response]

Thank you for pointing this out. We have changed the relevant part to the appropriate text.

  1. In Introduction, please add also that lung cancer is the leading cause of death in JApan and Worldwide!

[response]

Thank you for pointing this out. I have added that this is a problem in Japan as well as worldwide.

  1. In introduction, line 69-70...a rapid and highly diagnostic quality...should be rephrased...maybe change to ...a rapid and correct diagnosis in biopsy or surgically removed....

[response]

Thank you for pointing this out. We have changed the relevant part to the appropriate text.

  1. In the introduction, line 84, please change to...In this study, we found that Musashi-1 is strongly and diffusely expressed in LCNEC and SCLC.

[response]

Thank you for pointing this out. We have changed the relevant part to the appropriate text.

  1. Last words in the description of Figure 2...is it correct Figure 1???

[response]

Thank you for pointing this out. When rearranging the images, I had missed changing the images of specimens that did not express CD56 from l to o. I have corrected the relevant part.

  1. In Figure 3, I would suggest the full name and not MSI-1, or adding a legend?

[response]

Thanks for the suggestion, I changed the notation of Musashi-1 in figure 3 to full name.